# Microstructural profiles of the human superficial white matter and their associations to cortical geometry and connectivity

Youngeun Hwang[1,2*], Raul Rodriguez-Cruces[1,2], Jordan DeKraker[1,2],
Donna Gift Cabalo[1,2], Ilana R. Leppert[2], Risavarshni Thevakumaran[2], Christine L. Tardif[2,3],
David A. Rudko[2,3], Casey Paquola[4], Pierre-Louis Bazin[5], Andrea Bernasconi[2],
Neda Bernasconi[2], Luis Concha[6], Alan C. Evans[2], Boris C. Bernhardt[1,2*]

1 Multimodal Imaging and Connectome Analysis (MICA) Lab and Centre for Excellence in Epilepsy at the
Neuro (CEEN), Montreal Neurological Institute and Hospital, McGill University, Montreal, Québec, Canada,
2 McConnell Brain Imaging Centre, Montreal Neurological Institute and Hospital, McGill University,
Montreal, Québec, Canada, 3 Department of Biomedical Engineering, McGill University, Montreal,
Quebec, Canada 4 Institute for Neuroscience and Medicine (INM-7), Forschungszentrum Juelich, Juelich,
Germany, 5 Full Brain Picture Analytics, Leiden, the Netherlands, 6 Institute of Neurobiology, Universidad
Nacional Autónoma de México (UNAM), Querétaro, Querétaro, Mexico

* youngeun.hwang2@mail.mcgill.ca (YH); boris.bernhardt@mcgill.ca (BCB)

Medical Center Hamburg-Eppendorf:
Universitatsklinikum Hamburg-Eppendorf,
GERMANY

**Peer Review History:** PLOS recognizes the
benefits of transparency in the peer review
process; therefore, we enable the publication
of all of the content of peer review and
author responses alongside final, published
articles. The editorial history of this article is
available here: https://doi.org/10.1371/journal.
pbio.3003629

## Abstract

The superficial white matter (SWM), immediately beneath the cortical mantle, is
thought to play a major role in cortico-cortical connectivity as well as large-scale
brain function. Yet, this compartment remains rarely studied due to its complex
organization. Our objectives were to develop and disseminate a robust computa-
tional framework to study SWM organization based on 3D histology and high-field
7T MRI. Using data from the BigBrain and Ahead 3D histology initiatives, we first
interrogated variations in cell staining intensities across different cortical regions and
different SWM depths. These findings were then translated to in vivo 7T quantitative
myelin-sensitive MRI, including T1 relaxometry (T1 map) and magnetization trans-
fer saturation (MTsat). As indicated by the statistical moments of the SWM intensity
profiles, the first 2 mm below the cortico-subcortical boundary were characterized by
high structural complexity. We quantified SWM microstructural variation using a non-
linear dimensionality reduction method and examined the relationship of the resulting
microstructural gradients with indices of cortical geometry, as well as structural and
functional connectivity. Our results showed correlations between SWM microstruc-
tural gradients, as well as curvature and cortico-cortical functional connectivity. Our
study provides novel insights into the organization of SWM in the human brain and
underscores the potential of SWM mapping to advance fundamental and applied
neuroscience research.

**Data availability statement:** BigBrain histological data are accessible at https://bigbrain.loris.ca. AHEAD brain histology data utilized in this study can be found at https://doi.org/10.21942/uva.16844500.v1. The 7T data are available on the Open Science Framework at https://osf.io/mhq3f/. The code for SWM surface sampling associated with this project is provided in our open repository at https://doi.org/10.5281/zenodo.11510179. The analysis code and notebooks are accessible on GitHub (https://github.com/MICA-MNI/MRI-Histology_framework_for_SWM_architecture), and a versioned, citable copy of the code is available through Zenodo (https://doi.org/10.5281/zenodo.17896638). All datasets used for the analyses are hosted on the Open Science Framework at https://osf.io/e6f7d/.

**Funding:** Y.H. is funded by the Helmholtz International BigBrain Analytics and Learning Laboratory (HIBALL, https://hiball.org) and the Quebec BioImaging Network (QBIN, https://www.rbiq-qbin.qc.ca/). R.R.C is funded by the Fonds de la Recherche du Québec – Santé (FRQ-S, https://frq.gouv.qc.ca/en/health/), the Montreal Neurological Institute Jeanne Timmins Costello Fellowship (https://www.mcgill.ca/neuro/), and the Healthy Brains, Healthy Lives – Entrepreneur Postdoctoral Fellowship (https://www.mcgill.ca/hbhl/). J.D. is funded by the Natural Sciences and Engineering Research Council of Canada Postdoctoral Fellowship (NSERC-PDF, https://www.nserc-crsng.gc.ca/). D.G.C. is funded by the Fonds de la Recherche du Québec – Santé (FRQ-S, https://frq.gouv.qc.ca/en/health/) and the Savoy Foundation (https://www.savoy-foundation.ca/). C.T. is funded by FRQ-S (Research scholar J2) and Killam Trusts. L.C. is funded by the Consejo Nacional de Ciencia y Tecnología (CONACYT, https://www.conacyt.mx/) (181508, 1782, FC218-2023) and Dirección General de Asuntos del Personal Académico (DGAPA, https://www.unam.mx/) (IB201712, IG200117, IN204720, IN213423). B.C.B. acknowledges support from the Natural Sciences and Engineering Research Council of Canada (NSERC Discovery, https://www.nserc-crsng.gc.ca/), the Canadian Institutes of Health Research (CIHR, https://cihr-irsc.gc.ca/), the SickKids Foundation (https://www.sickkidsfoundation.

## Introduction

The superficial white matter (SWM) is the area of white matter located immediately underneath the cortical gray matter. Its structural and functional properties differ significantly from those of deep white matter, largely due to its close association with the overlying cortex. Within the SWM, short association fibers known as U-fibers interconnect adjacent brain gyri [1] and contribute significantly to the overall volume of white matter in humans [2], playing a pivotal role in brain plasticity and aging [3]. These fibers remain incompletely myelinated until later in life [1] and exhibit alterations in density in various neurological disorders, including autism [4], epilepsy [5], and Alzheimer's disease [6]. In addition to U-fibers and the proximal and distal ends of long-range association fibers, the SWM is characterized by a high density of white matter interstitial neurons [7], which are unevenly distributed across regions. Historically, the SWM was regarded as a passive conduit for information transfer. However, the presence of these neurons suggests a more dynamic and complex role [8–11]. Neural circuits within the SWM have the potential to modulate cortico-cortical connectivity by regulating the timing and efficiency of signal transmission at axonal connections [12,13]. This implies that the SWM is not merely a passive signal relay, but rather a critical modulator of brain connectivity and large-scale brain function.

Despite its biological and clinical significance, the SWM has remained understudied, primarily due to technical difficulties in mapping its complex anatomy [14]. *Postmortem* histological studies provide direct insights into brain microstructure and cellular microstructure, but there is a limited amount of high-resolution 3D datasets available. Recent advancements in neuroimaging, particularly high-field 7 Tesla (T) MRI, enable precise, noninvasive imaging and mapping of brain microstructure, including the thin SWM, across multiple individuals. Myelin-sensitive imaging techniques, for example, can differentiate regions with distinct myeloarchitectural profiles at an individual level while resolving complex fiber configurations previously inaccessible with standard imaging [15–19]. Notably, the versatility of MRI allows for the integration of multiple imaging contrasts, which opens new avenues for understanding the structural and functional role of SWM in the living brain. Resting-state functional MRI, for example, enables the identification of intrinsic networks [20–25], thereby facilitating the exploration of associations between SWM structure and macroscale brain function. Indeed, there is already evidence from previous lesion studies [5,26] showing that atypical SWM organization relates to altered functional connectivity patterns, which motivates further research on how variations in SWM microstructure affect large-scale brain function.

In this study, we aimed to develop a framework for densely mapping the microstructural organization of the SWM. Specifically, we designed an open-source surface-based pipeline in Python that enables the systematic sampling of SWM surfaces and microstructural characteristics on both 3D histology and high-field MRI datasets. First, we analyzed the microstructural complexity of the SMW using *postmortem* 3D histology to identify key features of cellular architecture and validate our surface-based sampling methodology. We translated this methodology to in-vivo 7T quantitative MRI to interrogate myeloarchitectural SWM profiles and establish spatial

com/), HIBALL, Healthy Brains, Healthy Lives (HBHL, https://www.mcgill.ca/hbhl/), Brain Canada Foundation (https://braincanada.ca/), FRQS(https://frq.gouv.qc.ca/en/health/), the Canada Research Chairs Program (https://www.chairs-chaires.gc.ca/), and the Centre for Excellence in Epilepsy at The Neuro (CEEN). The funders had no role in study design, data collection and analysis, decision to publish, or preparation of the manuscript.

**Competing interests:** P.B. is owner of Full brain picture Analytics. The authors have declared that no competing interests exist.

**Abbreviations:** CSF/GM, cerebrospinal fluid/gray matter; GM/WM, gray matter/white matter; MPC, microstructural profile covariance; MT, magnetic transfer; MTsat, magnetization transfer saturation; SD, standard deviation; SWM, superficial white matter, T1w, T1-weighted.

correspondence between in-vivo and *postmortem* information. Next, we examined the relationship between SWM organization, cortical geometry, as well as functional and structural connectivity. Lastly, we make all codes and normative SWM feature maps openly available to facilitate rapid adoption and independent verification of our approach.

## Results

### 1. Data-driven histological analysis of the human SWM

To systematically characterize the cellular organization of the SWM, we first examined histological intensity profiles derived from *postmortem* 3D histology. Using a surface-based sampling approach, we quantified the depth-dependent variation of cellular and fiber architecture within the SWM (Fig 1A). SWM microstructural intensity profiles were generated for each histological feature by sampling staining intensities along 50 equipotential surfaces, ranging from the gray matter/white matter (GM/WM) interface up to 3 mm into the white matter (Fig 1B). These profiles represent the staining intensities of cell bodies (BigBrain—Nissl), neuronal fibers (Ahead—Bielschowsky), and parvalbumin interneurons (Ahead—immunoreactivity) at varying SWM depths. Near the GM/WM boundary, the staining intensities for cell bodies and neuronal fibers tend to increase, while staining for parvalbumin interneurons shows a decreasing trend. Cellular profiles in the SWM markedly varied depending on the cytoarchitectural type of the overlying cortex, especially in the upper portion of the SWM (Fig 1C). Qualitative assessments indicated that the strongest variation in cell body intensity within the SWM was observed in idiotypic regions with marked laminar differentiation, while the most notable variation in neuronal fiber intensity was found in paralimbic regions with only subtle laminar differentiation and in the insula. Notably, and as expected, across all histological intensity profiles, the most significant changes were observed between 0 and 1 mm below the GM/WM interface.

### 2. Data-driven MRI analysis of the human SWM

To extend our histology-based analysis to in vivo data, we translated the same SWM surface-based sampling methodology for use with 7T MRI, which allowed us to generate SWM microstructural intensity profiles from the T1 and MTsat maps of individual subjects (Fig 2A). These profiles were created by sampling MRI intensities along 15 equipotential surfaces extending from the GM/WM interface to 3 mm into the white matter (Fig 2B). Each profile represents the intensity of myelin-sensitive MRI measures, T1 map, and MTsat. The T1 map showed a gradual decrease in intensity, while MTsat exhibited a corresponding increase, with the progressive rise in myelin content in deeper regions. Qualitative evaluations of laminar differentiation and cytoarchitectural complexity (Fig 2C) demonstrated notable SWM intensity variations, particularly within paralimbic and insular regions. In those areas, both T1 map and MTsat profiles demonstrated pronounced intensity shifts between 0 and 2 mm beneath the GM/WM interface, highlighting the dynamic nature of microstructural properties in this depth range.

**A | Sampling SWM microstructure from histology**

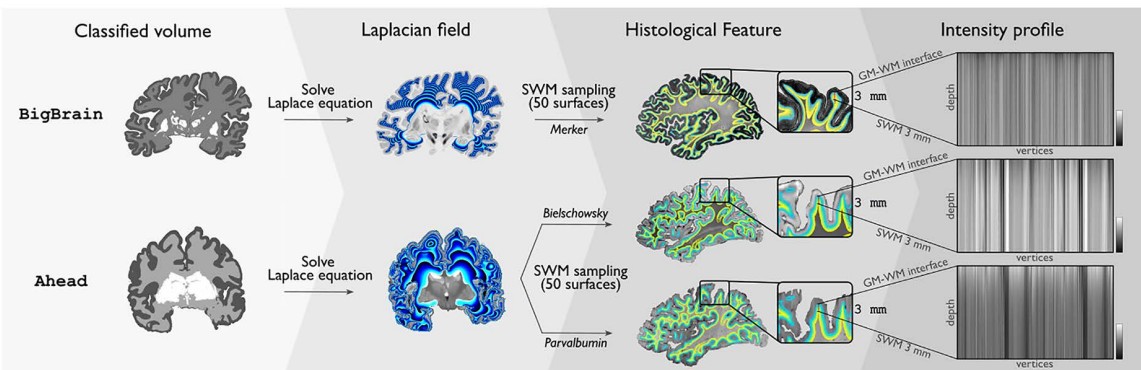

**B | Intensity profiles across WM depths derived from each histological dataset**

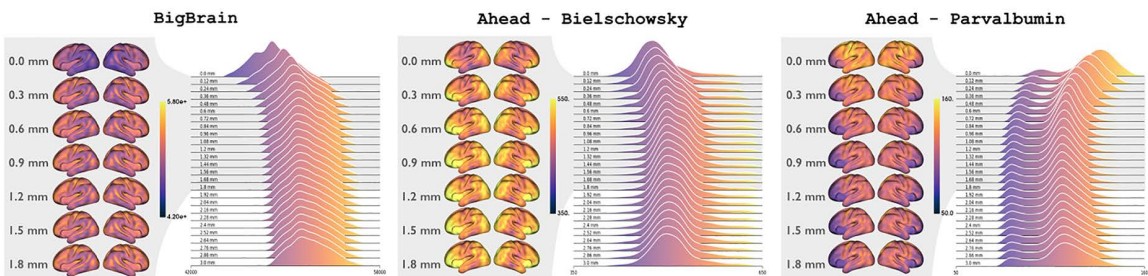

**C | Cellular intensity profiles across SWM depths**

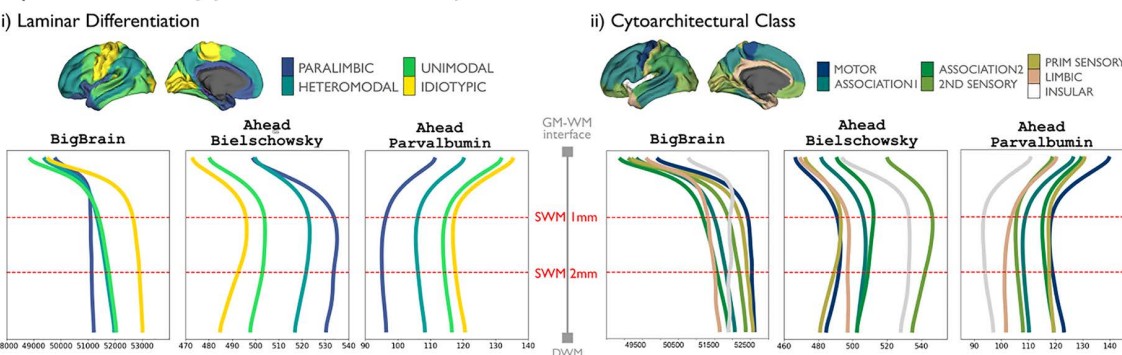

**Fig 1. Histology-based SWM microstructure intensity profile. (A)** SWM sampling framework from 3D *postmortem* histology. The Laplacian field within the white matter domain was solved for each histological classified volume to sample 50 SWM surfaces, spanning from the GM/WM interface to 3 mm into the white matter. These surfaces were then mapped to each histological staining method, allowing the extraction of intensity profiles for each vertex across SWM depths. **(B)** Alterations in the intensity distributions of histological features across SWM depths. The GM/WM interface (0 mm) and 25 representative SWM surfaces (up to 3 mm into the white matter) are shown. The full set of 50 SWM surfaces, and their intensity alterations are provided in S1 Fig. The surfaces showing the greatest variation in SWM, from 0 to 1.8 mm into the white matter, were mapped onto the brain alongside the intensity profiles on the *left*. **(C)** Spatial associations between the intensity profiles and levels of laminar differentiation (*left*) and cytoarchitectural class (*right*). See also S2 and S3 Figs for extended analyses of intensity distribution alterations across cortical types.

## 3. Comparative analysis of statistical moments across gray matter-white matter depths

To investigate intensity changes from cortical areas into the white matter, we first obtained intensity profiles from the cerebrospinal fluid/gray matter (CSF/GM) boundary through the GM/WM interface and extending up to 3 mm into the SWM. At each depth, we then calculated central statistical moments, including mean, standard deviation (SD), skewness, and

**A | Sampling SWM microstructure from quantitative MRI**

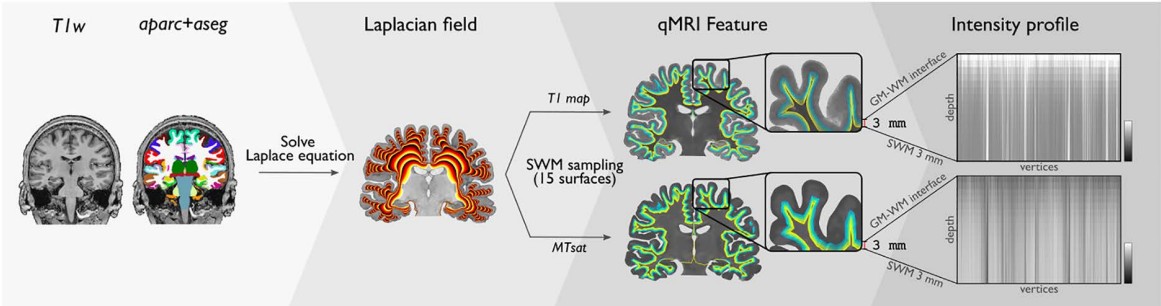

**B | Intensity profiles across WM depths derived from each quantitative MRI**

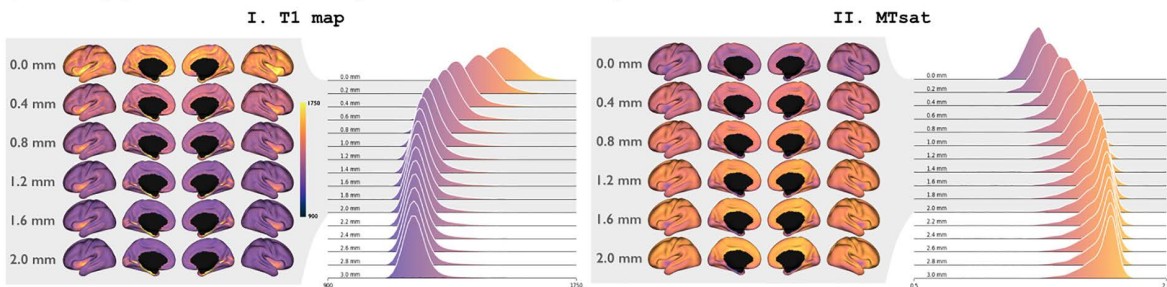

**C | Cellular intensity profiles across SWM depths**

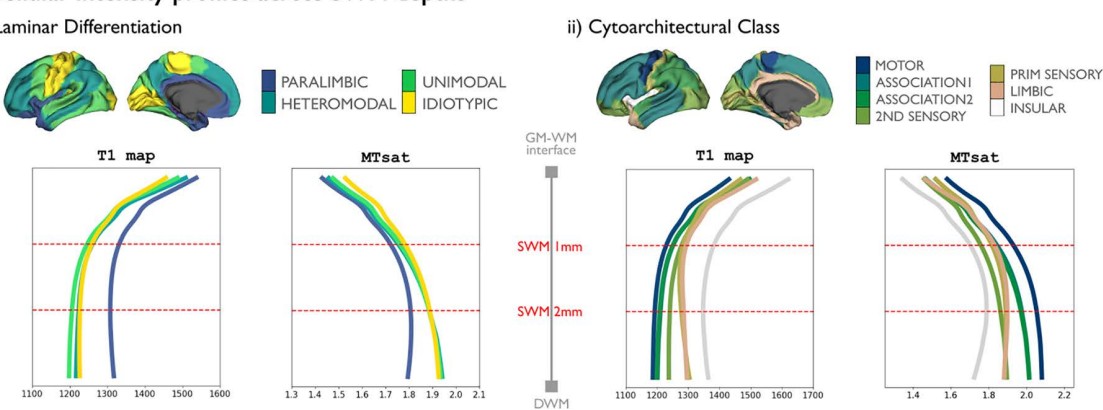

**Fig 2. In-vivo-based SWM microstructure intensity profile. (A)** SWM sampling framework from quantitative MRI. The Laplacian field within the white matter domain was solved for "*aparc+aseg*" file from each T1w image to sample 15 SWM surfaces, spanning from the GM/WM interface to 3 mm into the white matter. These surfaces were then mapped to each MRI, allowing the extraction of intensity profiles for each vertex across SWM depths. **(B)** Alterations in the intensity distributions of MRI features across SWM depths. The GM/WM interface (0 mm) and 15 SWM surfaces (up to 3 mm into the white matter) are shown. The surfaces showing the greatest variation in SWM, from 0 to 2 mm into the white matter, were mapped onto the brain alongside the intensity profiles on the *left*. **(C)** Spatial associations between intensity profiles and levels of laminar differentiation (*left*) and cytoarchitectural class (*right*). See also S2 and S3 Figs for extended analyses of intensity distribution alterations across cortical types.

kurtosis, across the surface to quantify regional variation in microstructural properties. These features have typically been computed across cortical depths and linked to microstructural differentiation in classic and contemporary neuroanatomical studies [27]. By extending this approach to the SWM, we aimed to characterize how intensity values at each depth vary across regions, thereby capturing spatial heterogeneity in microstructure across white matter depths and offering a more comprehensive understanding of the SWM's structural complexity.

Histological and in-vivo intensity profiles revealed microstructural complexity across the SWM, particularly within the first 2 mm of the white matter (Fig 3). Histological profiles (Fig 3A) typically exhibited peak variability within the gray matter and revealed a distinct transition zone near the GM/WM interface (marked by the red line). Notably, within the white matter, all histological markers displayed pronounced intensity changes within approximately the first 1 mm, irrespective of variations in the overlying cortex. In the in-vivo MRI data, the T1 map (Fig 3B(i)) displayed a gradual decrease in intensity from gray matter

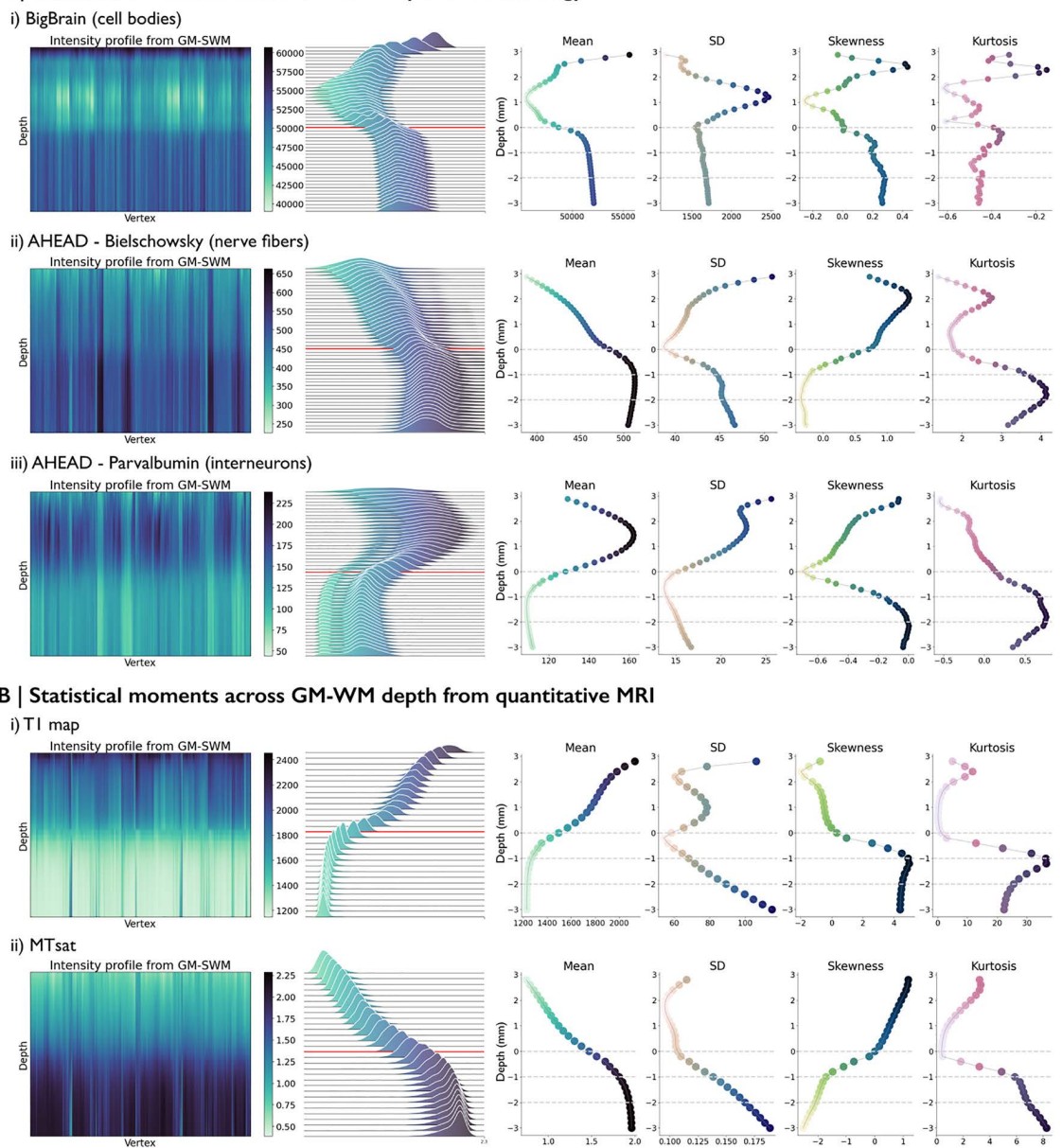

**Fig 3. Microstructural profiling of SWM across (A) histology and (B) MRI datasets.** Vertex-wise microstructural intensity profiles are shown on the *left*. Intensity distributions by gray matter-white matter depth, with the GM/WM interface marked by a red line, are displayed in the *middle*. Statistical moments across the cortex and SWM are shown on the *right*. Gray dotted lines indicate every 1 mm within the SWM.

to white matter, with pronounced shifts near the GM/WM interface, aligning with increased myelin content. Similarly, MTsat (Fig 3B(ii)) revealed a steep increase in intensity within the first 1 mm of white matter, reflecting its sensitivity to myelin-related transitions. In addition to mean intensity, statistical moments (SD, skewness, and kurtosis) further captured regional complexity across depths. The consistently low SD observed at the GM/WM interface across most modalities, with the exception of MTsat, likely reflects a shared anatomical transition across cortical areas, where similar intensity changes yield reduced regional variability. In contrast, the heightened variability from the GM/WM interface to 2 mm into the white matter indicates a structured yet dynamic progression of microstructural properties across white matter depths.

Together, these findings indicate that SWM is not a homogeneous region but a structured continuum, exhibiting depth-dependent microstructural variations shaped by cellular, fiber, and myelin organization, which vary across the brain. Histological data captured cellular differentiation, while myelin-sensitive MRI provided broader tissue contrasts extending deeper into white matter, offering complementary insights into the microstructural organization of the SWM.

## 4. Cortical geometry and MRI microstructural gradient associations in SWM

As spatial patterns of SWM microstructure were revealed, we examined their relationship to cortical geometry. A nonlinear dimensionality reduction approach provides a low-dimensional representation of spatial organization, allowing us to capture continuous axes of variation that may reflect fundamental principles of SWM architecture. Using microstructural gradients, we derived gradients that capture spatial pattern of SWM microstructural similarity (Fig 4A). The primary gradient (G1) explained 57% of the variance in the T1 map and 52% in MTsat, highlighting its dominant contribution relative to higher-order gradients (S4 Fig). Visual inspection of these spatial patterns suggests that the primary gradient derived from the T1 map varies along the main sulci and gyri, while the gradient derived from MTsat appears anchored at one end by the central sulcus, the insula, and the frontal blade [28] (Fig 4A, *right)*. The stability of group principal gradient was quantified by calculating the Pearson correlation coefficients between the spatial gradients of individual subjects, yielding $r = 0.37$ for the T1 map and $r = 0.01$ for MTsat. All following analyses using the gradients were performed individually for each subject. For reference, we also computed histology-based gradients, which are provided in S5 Fig.

To assess the association between brain geometry and SWM intensity variation, we examined the correlation between the principal gradients of each MRI and cortical features at various SWM depths (Fig 4B). Curvature, as it varies across gyral and sulcal regions, showed a strong correlation with $G1_{T1map}$ on the global surface (mean $r = 0.33$ across subjects, $P_{spin} < 0.05$ for all subjects) (Fig 4B(i)). Compared to the T1 map, $G1_{MTsat}$ exhibited weaker but still significant correlations with curvature (mean $r = 0.15$ across subjects, $P_{spin} < 0.05$ for all subjects). Within the gyral and sulcal surfaces separately, correlations in both MRI measures were weaker than those observed on the global surface. Cortical thickness (Fig 4B(ii)) demonstrated a moderate correlation with $G1_{T1map}$ (mean $r = -0.17$ across subjects, $P_{spin} < 0.05$ for 9 subjects) but showed no significant association with $G1_{MTsat}$.

These findings suggest that variations in T1 map intensity within the SWM are partially related to cortical geometry, with cortical folding patterns relating to intensity distributions.

## 5. SWM microstructural gradients correlations with functional and structural connectivity

To further understand the organizational role of SWM, we investigated whether its microstructural gradients reflect the large-scale connectivity patterns in the brain. Specifically, we examined the relationship between SWM gradients and both functional and structural connectivity, to explore their potential role in supporting cortical communication (Fig 5).

First, we examined the correlation between the principal gradient of each MRI and functional strength (Fig 5A). Higher functional strength indicates that a region actively participates in information exchange. Vertex-wise correlation analysis revealed a significant association between functional strength and $G1_{T1map}$ in the group analysis ($r = -0.24$, $P_{spin} = 0.001$). In subject-level analyses, this association was consistent, with a mean correlation of $r = -0.17$ ($P_{spin} < 0.05$ for all subjects).

Structural strength (Fig 5B), which reflects the degree of neuronal fiber connectivity within the cortex, demonstrated a stronger correlation with $G1_{T1map}$ in the group analysis ($r = -0.38$, $P_{spin} < 0.05$). In subject-level analyses, this relationship

## A | Constructing microstructural gradients in SWM

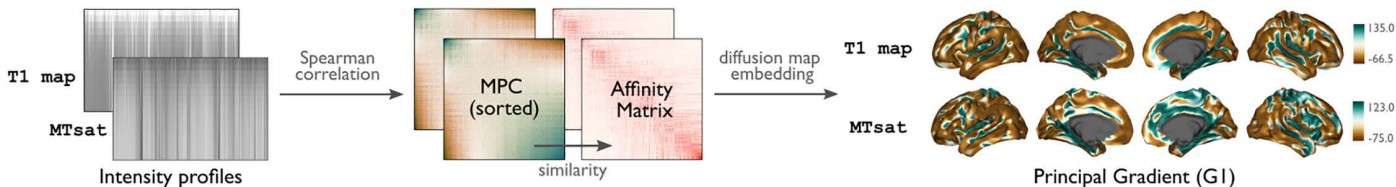

## B | Cortical geometry correlations of qMRI gradients

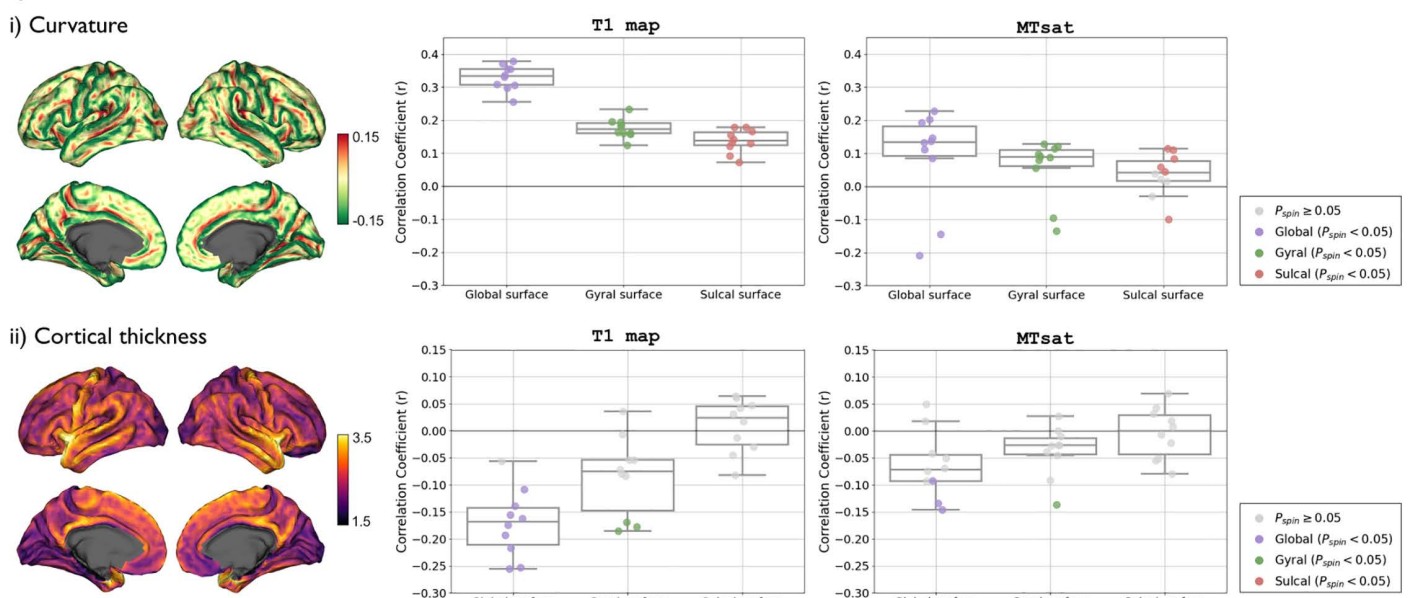

**Fig 4. SWM microstructural gradients and their correlation with cortical geometry. (A)** MPC matrices were individually computed from intensity profiles derived from each MRI. The transformed affinity matrices were then subjected to diffusion map embedding to obtain SWM microstructural gradients for each MRI. **(B)** Subject-wise correlations between cortical geometry and MRI gradients. Box-and-whisker plots illustrate the distribution of Spearman correlation coefficients across 10 subjects. The data underlying this figure can be found in S1 Data. Data points with nonsignificant correlations are marked as light gray dots. Additionally, correlations were analyzed separately for gyral and sulcal vertices. **Abbreviations**: $r$, Spearman correlation; $P_{spin}$, $p$-value as determined by a spin test that controls for spatial autocorrelation.

was also evident, with a mean correlation of $r = -0.23$ across subjects ($P_{spin} < 0.05$ for all subjects). By contrast, $G1_{MTsat}$ showed a correlation with structural strength in the group analysis ($r = -0.312$, $P_{spin} = 0.024$), and most individual subjects did not show significant correlations (mean $r = -0.025$ across subjects, $P_{spin} < 0.05$ for 6 subjects).

Overall, $G1_{T1map}$ demonstrated consistent and significant correlations across both group- and subject-level (see S6 Fig for the preliminary short-range structural connectivity analysis), while $G1_{MTsat}$ results showed weaker and less consistent correlations. These findings indicate that T1 map intensity variations within the SWM are moderate related to both functional and structural connectivity, support a potential role in understanding brain organization.

## Discussion

In this study, we investigated the microstructural integrity of the SWM using 3D *postmortem* histology and in-vivo 7T MRI, establishing an openly available framework for SWM exploration. This framework provides a surface-based, reproducible method for sampling SWM intensity profiles and microstructural covariance, while requiring minimal dependencies to

## A | Functional connectivity strength correlates of qMRI gradients

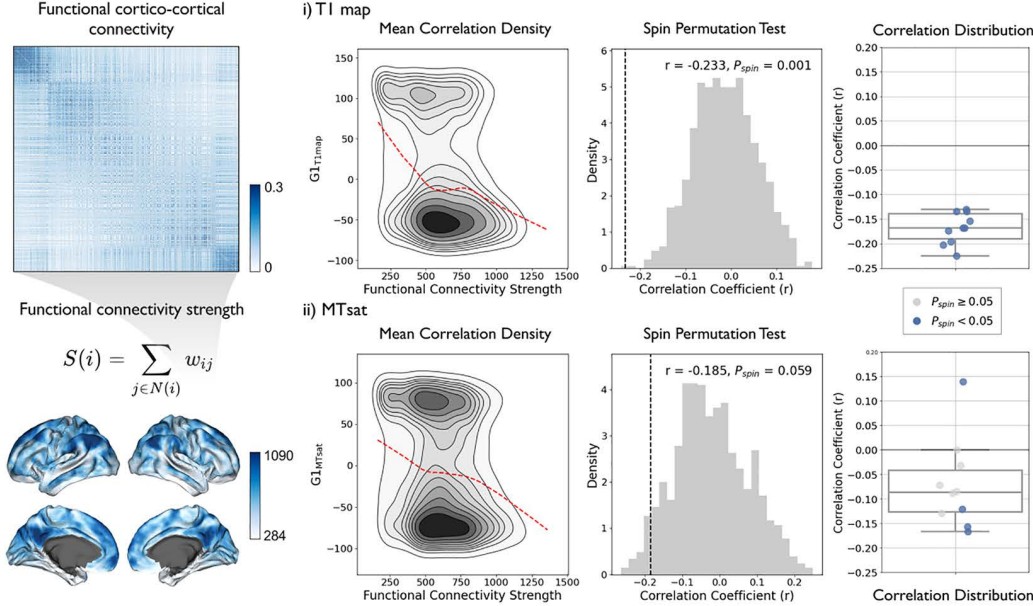

## B | Structural connectivity strength correlates of qMRI gradients

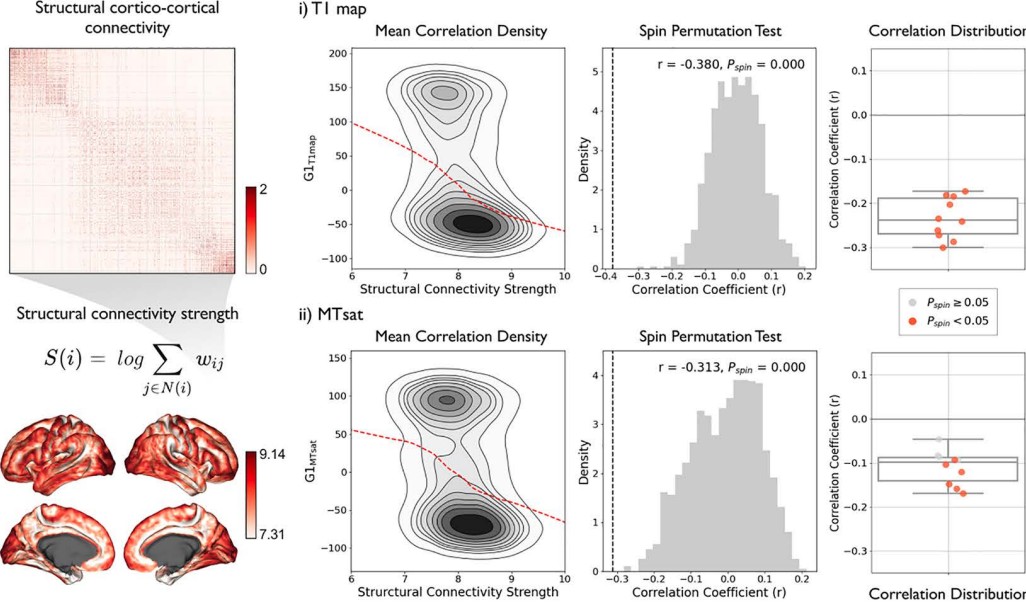

**Fig 5. Functional connectivity (FC) and structural connectivity (SC) correlation analysis with SWM gradients. (A)** FC and **(B)** SC strength were calculated as the weighted sum of the connectivity matrix across all nodes (*left* panel). Matrices were sorted by regions with greater similarity and nonlinearity. The *middle* panel shows the density plot of nonlinear correlations between connectivity strength and SWM gradients in the group analysis, along with statistical significance assessed using a spin permutation test. The red dotted line in the density plots represents a nonlinear fit capturing the relationship between the gradients and the connectivity strengths. The *right* panel presents subject-wise correlations between each connectivity strength and the SWM gradients from each qMRI. Box-and-whisker plots illustrate the distribution of Spearman correlation coefficients across 10 subjects, with nonsignificant correlations shown as light gray dots. The data underlying this figure can be found in S1 Data. **Abbreviations**: *r*, Spearman correlation; $P_{spin}$, Significance.

ensure wide accessibility. By analyzing intensity profiles and statistical moments across different SWM depths, we identified pronounced structural complexity within the first 2 mm of the SWM, reflecting significant variations in histological staining and myelin-sensitive MRI intensities. This suggests that the gradual changes seen across depths primarily reflect genuine microstructural variation rather than imaging artifacts and are consistent with significant variations in histological staining and myelin-sensitive MRI intensities. While depth-specific intensity profiles provide valuable information, they capture microstructural variations only at discrete sampling points. To overcome this limitation, we applied a diffusion map embedding approach, which allows identification of continuous gradients of microstructural similarity across the SWM depths. This approach captures the primary spatial axes of variance in SWM microstructural profiles, revealing systematic variations linked to cortical geometry, as well as both structural and functional connectivity. Notably, SWM microstructural gradients derived from T1 maps correlated with both structural and functional connectivity, reinforcing the idea that SWM organization is closely tied to large-scale brain networks. These findings highlight the intricate nature of SWM and provide a foundation for its potential relevance in neuroimaging studies, with implications that may extend to clinical contexts.

Historically, the structural complexity of the SWM has challenged its neural characterization [13,29]. However, systematic patterns of variability across both gray and white matter regions have been shown to facilitate the reliable characterization of cortical and subcortical areas [30–32]. By utilizing multiple histological stainings, we have advanced the understanding of the SWM's microstructure, revealing substantial changes in mean intensity within the first 1 mm into the white matter, particularly in cell body and interneuron stainings. Given that histological data provide high-resolution insights into cellular features, their staining properties are often chosen to emphasize cortical regions. Yet, these findings highlight the importance of understanding the transitional zones at the GM/WM interface, where diverse cellular and structural properties shape connectivity patterns. From a cytoarchitectonic perspective, this transition is further highlighted by differences observed across species. In rodents, the deepest cortical layer consists of a thin layer of persistent subplate cells overlying a myelin-rich, cell-sparse stratum through which many corticocortical axons traverse [33]. Neurons in this layer contribute to both local and long-distance corticocortical connections. In contrast, in the human brain, the deepest cortical layer (layer VI) is primarily composed of fusiform cells, with pyramidal cells and interneurons playing less prominent roles [34]. Beneath this lies the white matter, where white matter interstitial neurons are located directly underneath the cortical ribbon. These neurons have been shown to form cortico-cortical and cortico-thalamic connections, including in the human supratemporal plane [35]. These white matter interstitial neurons play a critical role during cortical development [36,37] and have been implicated in psychiatric disorders [38–41], with autism in particular showing alterations in their density and distribution [42,43].

The myeloarchitecture of the SWM represents another key aspect of its organization. The SWM comprises a diverse mix of fiber populations with varying orientations and destinations, creating a complex 3D structure of fiber bundles [44]. Local differences in the thickness and compactness of fiber layers, as well as the length and density of radial bundles, enable the identification of distinct myeloarchitectonic regions [45,46]. Building on this, our study translated myelin-sensitive MRI to investigate depth-wise surface-based properties spanning from the cortical regions into the SWM in the human brain. The findings revealed that, across all modalities, mean intensity changed substantially within the first 1 mm, while statistical moments continued to reflect sustained regional variability up to a depth of 2 mm. This indicated that the first 2 mm of the SWM region is characterized by high structural complexity, likely reflecting the interplay between tangentially oriented U-fibers, which run parallel to the GM/WM interface, and radially aligned cortical afferents and efferents, which intersect perpendicularly [47,48]. Beyond approximately 2 mm into the white matter, in most regions, structural complexity appeared to stabilize, consistent with a transition to more homogeneous long fiber bundles characteristic of the deep white matter. Our findings support the notion that the SWM forms a structured continuum rather that a homogeneous region, with depth-dependent microstructural variations shaped by cellular, fiber, and myelin organization across different brain areas, supporting these organizational principles. Similar depth-dependent patterns have also been observed across white matter regions in nonhuman primates [32,49–51]. While this study does not explicitly disentangle the contributions

of these fiber systems, future investigations using histological and diffusion MRI techniques could help delineate their respective roles in shaping SWM architecture.

Our understanding of the human SWM was expanded through microstructural gradient approaches that capture microstructural variation within the SWM [5]. The primary gradient derived from the T1 map clearly differentiates between the main sulci and gyri, implying that their intensity changes are associated with brain folding [52], which also showed a significant correlation with brain curvature. Our findings also demonstrated a correlation between the T1 map gradient and cortico-cortical functional and structural connectivity in the SWM region. To bridge microstructural and macroscale perspectives, we examined how the principal gradient of SWM microstructure relates to regional connectivity strength. As this measure reflects each region's integration within the global connectome, our underlying hypothesis was that variations along the gradient may correspond to distinct roles in both local and large-scale network communication. This approach allows us to probe the interplay between microstructural organization and macroscale connectivity patterns. This effect was more robust for the T1 map than for MTsat. One possible explanation is that the T1 map has higher spatial resolution and is sensitive to a broader range of microstructural features, including iron, which may be particularly relevant for SWM organization. Specifically, the T1 map showed a stronger correlation with structural connectivity, which may reflect the contribution of U-fibers, known to support local cortical communication and as a predominant feature of the SWM [53]. This relationship suggests that intensity variation could be used to characterize pathological alterations in the SWM associated with various diseases. In particular, SWM characterization could be applied to better understand both typical as well as atypical brain development, and to clarify microarchitectural mechanisms contributing to neurological conditions such as focal cortical dysplasia [54,55] as well as developmental psychopathologies [55,56]. Considering the general assumption that structural connections largely determine functional dynamics [57], we would nevertheless highlight the need for further investigation of subcortical U-fibers. Combining with our profiling method, diffusion MRI-derived tractography could serve as a proxy for U-fibers, informed by local cyto- and myelo- architecture, to further elucidate the connectivity and functional significance of this intricate brain region. While this study focused on characterizing surface-based SWM properties, analyzing short-range structural connectivity indicated that the SWM properties retain their relationship with local connectivity patterns. Building on this, future research will incorporate tractography-based approaches to directly investigate the organization and functional role of short-range, U-fiber bundles in SWM connectivity.

These insights underscore the potential of SWM mapping as a valuable biomarker for advancing research and clinical applications. Our approach emphasizes open science in SWM research, as demonstrated by the development of an open-access repository for sampling SWM surfaces, the use of publicly available datasets, and the sharing of findings to promote transparency and collaboration. By validating SWM mapping with both histological and in-vivo MRI datasets, we established a robust and reproducible framework for studying this intricate brain region. Furthermore, the integration of curvature-corrected SWM intensity profiles and functional MRI data enhances our understanding of SWM's functional role in living humans. Through these efforts, we have laid the groundwork for an open-science framework in SWM research, providing tools and methodologies that encourage collaboration and drive advancements in precision medicine and neuroscience.

## Materials and methods

### 1. Histological data acquisition and preprocessing

We utilized two open-access 3D histology datasets: BigBrain [58] and Ahead brain [59].

**a. BigBrain acquisition and preprocessing.** The BigBrain dataset [58] is an ultra-high-resolution Merker-stained 3D volumetric histological reconstruction of a *postmortem* human brain from a 65-year-old male, available through the open-access BigBrain repository (https://bigbrain.loris.ca). The brain was embedded in paraffin, sectioned coronally into 7,400 slices at 20 μm thickness, silver-stained to highlight cell bodies [60], and digitized. After manual artifact inspection, automated repair processes were applied, including nonlinear alignment to a *postmortem* MRI, intensity normalization,

 

and block averaging [61]. The 3D reconstruction was completed using a hierarchical coarse-to-fine hierarchical procedure [62]. Geometric meshes approximating the CSF/GM and the GM/WM boundaries, each aligned to the Conte69 template (164k vertices per hemisphere), were also available in the repository. Histological staining intensity profiles in the cortical area are openly accessible as part of BigBrainWarp (http://github.com/caseypaquola/BigBrainWarp) [63]. These profiles include histological intensities sampled along 50 equivolumetric surfaces spanning from the CSF/GM to GM/WM boundaries, and the CSF/GM surface was removed to reduce the impact of partial volume effects.

For our analysis, we primarily used 200 μm resolution data to create classified volumes and sample the SWM surfaces, while 100 μm resolution images were subsequently used to map its microstructural properties onto these surfaces. To optimize computational efficiency, surfaces were downsampled to 5k vertices per hemisphere using *connectome* work-bench [64]. Additionally, outliers in the temporal pole region of the data were identified using Tukey's method, which calculates the interquartile range to define thresholds for detecting extreme values. These outliers were addressed through interpolation and extrapolation to ensure data completeness and reliability.

**b. Ahead brain acquisition and preprocessing.** The Ahead brain [59] open-access dataset [65] is a multimodal 3D reconstruction of the whole human brain at 200 μm resolution, created by integrating ultra-high-field MRI and light microscopy from two *postmortem* human brains. Quantitative MRI maps of relaxation rates (R1, R2*) and proton density were obtained by imaging the brain at 400 μm intervals [66,67]. Additionally, microscopic images were obtained coronally by sectioning the brain at 200 μm intervals and applying six different staining techniques. The ANTs SyN algorithm [68] was used to re-register blockface and microscopic images, minimizing nonlinear distortions and ensuring accurate alignment of brain boundaries and vascular maps. Although small intensity variations were present in the 3D reconstruction, these were linearly adjusted to preserve the original dynamic range of each stain. In our study, we utilized a single subject from the dataset, a 59-year-old female (https://doi.org/10.21942/uva.16844500.v1), and selected only Bielschowsky and Parvalbumin stainings, which target nerve fibers and interneurons, respectively, with a focus on white matter regions. Since the provided MRI and segmentation data exhibited artifacts in the occipital lobe, we used the blockface image with FastSurfer [69] to obtain the CSF/GM and GM/WM boundaries. As FastSurfer recommends a T1-weighted (T1w) image as input and has resolution limitations, particularly for histological datasets exceeding 0.7 mm isovoxel resolution [69]. Accordingly, the blockface image was downsampled to a 0.8 mm isovoxel resolution to generate a synthetic T1w image. This was achieved by inverting the downsampled blockface image, supplementing it with a synthetic head, and processing it to remove cerebrospinal fluid, which could interfere with cortical surface extraction. Additionally, nonlocal means denoising was applied to enhance contrast prior to running FastSurfer. As a result, we obtained classified volumes as well as CSF/GM and GM/WM surfaces. Subsequently, the original high-resolution images were used to map microstructural properties onto the SWM surfaces (S7 Fig). We also generated 50 equivolumetric surfaces ranging from the CSF/GM to GM/WM boundaries to sample histological intensities across cortical depths. This was achieved using openly available code (https://github.com/kwagstyl/surface_tools) [70] for generating equivolumetric surfaces, following methodologies previously validated for various in-vivo myelin-sensitive contrasts and a range of image resolutions [71]. This process produced distinct intensity profiles that reflect the intracortical microstructural composition at each cortical vertex. To minimize partial volume effects, data sampled at the CSF/GM boundaries were excluded. Similar to the BigBrain dataset, all the surfaces from this dataset were downsampled to 5k vertices per hemisphere using *connectome* workbench [64].

**c. Superficial white matter surface-based sampling and feature mapping from histology datasets.** To examine the SWM, we adopted a surface-based sampling approach, where the SWM surfaces represent computational reconstructions that facilitate the study of SWM in relation to the cortex [5,47]. We employed an equipotential-based method for SWM sampling, which offers advantages in smoothness and practical implementation while minimizing artifacts and the need for complex post-hoc corrections. Following this, we first solved the Laplace equation within the white matter domain. This was achieved by initially computing a Laplace field between GM/WM interface and

the ventricular walls and subsequently shifting an existing GM/WM surface along that gradient to this field. Stopping conditions were determined by the geodesic distance traveled. This potential field across the white matter guided the placement of SWM surfaces below the GM/WM interface. For the histological datasets, SWM surfaces were sampled at 50 depths, each spaced 0.06 mm apart beneath the GM/WM interface (Fig 1A). The maximum SWM depth of 3 mm was chosen to capture both the U-fiber system and the termination zones of long-range bundles [2]. All code used for SWM surface sampling is available in our open repository [72,73]. The Laplacian field provided an isomorphic mapping between points on the SWM surface and the overlying cortex, enabling seamless integration of gray matter and white matter metrics. These SWM surfaces were subsequently mapped to the original high-resolution space of each histological staining. Voxel-wise histological features were interpolated using trilinear interpolation to all surface points along the newly mapped SWM surfaces. Finally, the measurements were registered to the Conte69 template surface, which was downsampled to 5k vertices per hemisphere using *connectome* workbench [64]. We then characterized intensity profiles from the CSF/GM boundary to GM/WM interface and extending up to 3 mm into the WM using central moments of the intensity distribution, including mean, SD, skewness, and kurtosis, computed across the surface to quantify regional variation in microstructural properties.

## 2. MRI data acquisition and preprocessing

For the in-vivo dataset in the study, we utilized the ultra-high field imaging dataset, MICA-PNI [74]. A detailed imaging protocol is provided in the data release for additional reference (https://osf.io/mhq3f/ and https://portal.conp.ca).

   **a. In-vivo 7T MRI acquisition.** All 7T MR imaging data were acquired at the McConnell Brain Imaging Centre of the Montreal Neurological Institute. The dataset comprised 10 healthy participants (4 males, 6 females) with a mean ± SD age of 26.6 ± 4.60 years. All participants were recruited between March 2022 and November 2023. Multimodal MRI data were collected using a 7 Tesla (T) Siemens Terra system with a 32-channel receive and 8-channel transmit head coil operating in parallel transmission mode. For each MRI scan, the following parameters were used: (1) 3D-magnetization-prepared 2-rapid gradient-echo sequence [75] (MP2RAGE; 0.5 mm isovoxels, matrix = 320 × 320, 320 sagittal slices, TR = 5,170 ms, TE = 2.44 ms, TI1 = 1,000 ms, TI2 = 3,200 ms, flip = 4°, iPAT = 3, partial Fourier = 6/8, FOV = 260 × 260 mm$^2$) (2) Three images were acquired using the in-house magnetic transfer (MT) spoiled gradient echo sequence for MT saturation (MTsat) imaging: PD-weighted image (MT pulse off, 0.7 mm isovoxels, TR = 95 ms, flip angle = 5), MT-weighted image (MT pulse on, isovoxels, TR = 95 ms, flip angle = 5) and T1w image (MT pulse off, isovoxels, TR = 40 ms, flip angle = 25). (3) Multi-echo resting-state functional MRI with a 2D BOLD echo-planar imaging sequence [76] (University of Minnesota, CMRR; 1.9 mm isovoxels, 75 slices oriented to AC-PC-39 degrees, TR = 1,690 ms, TE = 10.80/27.3/43.8 ms, flip = 67°, FOV = 224 × 224 mm$^2$, slice thickness = 1.9 mm, MB = 3, echo spacing = 0.53 ms) (4) Diffusion-weighted MRI with three distinct shells at *b*-values of 0, 300, 700, and 2,000 s/mm$^2$. The number of diffusion directions for each shell was 10, 40, and 90, respectively (1.1 mm isovoxels, TR = 7,383 ms, TE = 70.60 ms, flip angle = 90°, refocusing flip angle = 180°, FOV = 211 × 211 mm$^2$, slice thickness = 1.1 mm, MB = 2, echo spacing = 0.79 ms)

   **b. Multimodal MRI Processing.** In this study, we utilized one structural MRI, two myelin-sensitive quantitative MRIs, functional MRI, and diffusion-weighted MRI from MICA-PNI [74]. The dataset included: (1) MP2RAGE-derived [77] T1w data deobliqued and reoriented to standard LPI orientation (left-to-right, posterior-to-anterior, and inferior-to-superior). (2) T1 map also obtained using the MP2RAGE sequence. Two inversion images were combined to create T1 maps, minimizing sensitivity to B1 inhomogeneities and enhancing reliability [77,78]. (3) MTsat [79,80], a semiquantitative myelin measure, was derived from MT imaging. Since traditional quantitative maps are susceptible to bias introduced by RF transmit field (B1+) inhomogeneities, all raw images underwent a unified segmentation-based correction [81] to address B1 + inhomogeneities. (4) Resting state functional MRI data were reoriented to LPI orientation. For multi-echo scans, time series were extracted from all echoes, nuisance signals were removed, echoes were optimally combined, and the resulting data were decomposed into components using principal and independent component analysis with TEDANA v.0.0.12 [82]. The data were subsequently subjected to

high-pass filtering and registered to the native cortical surface as volumetric time series. Functional connectivity matrices were computed for each individual using preprocessed resting-state functional MRI data. The time series were mapped to individual cortical surfaces and aligned to the standard template. Subsequently, subject-specific functional connectomes were constructed by cross-correlating the time series of all nodes, and the resulting correlation coefficients underwent Fisher's r-to-z transformation. In this study, we used functional cortico-cortical connectivity matrices based on the fsLR-5k symmetric surface template, resampled through subject-specific matrix generation. To assess functional connectivity strength, we focused on the absolute values of correlation coefficients, providing insight into the strength of connections regardless of direction [83]. Finally, functional strength, which reflects the overall connectivity of a given node, was calculated as the weighted sum of the functional connectivity matrix across all nodes. (5) Diffusion-weighted MRI data were preprocessed in native space using *MRtrix3* (https://mrtrix.readthedocs.io/en/latest/) [84], including Marchenko–Pasteur (MP-PCA) [85,86] denoising, Gibbs ringing correction [87,88], head motion and susceptibility distortion correction, eddy current correction, and non-uniformity bias field correction [88–91]. The b0 image was then extracted and aligned to the MP2RAGE-T1w structural image using linear registration. Fiber orientation distribution estimation was performed using the multi-shell multi-tissue constrained spherical deconvolution algorithm [92]. Subsequently, probabilistic whole-brain tractography was conducted using the iFOD2 algorithm with anatomically constrained tracking, followed by spherical-deconvolution informed filtering of tractograms strategy [93]. The relevant tracking parameters [94] were as follows: minimum/maximum track length = 10/400 mm, algorithm step size = 0.5 mm, and cutoff for track termination = 0.06. Structural connectivity matrices were generated based on the sum of streamlines, weighted by the respective node areas. Similar to functional connectivity, structural connectivity properties were assessed using vertex-wise strength based on the fsLR-5k symmetric surface. These were computed as the weighted sum of the subject-specific structural connectivity matrix across all nodes. Given the broad range of structural connectivity, we then applied the natural logarithm to structural connectivity strength to enhance interpretability.

Pre- and post-processing of these multimodal MRIs was performed using the open-access *micapipe* toolbox (version 0.2.3) (http://micapipe.readthedocs.io) [95]. All MRI images were aligned to native FastSurfer space of each participant using label-based affine registration. Subject-specific and quantitative image-specific series of 14 equivolumetric surfaces were generated between the CSF/GM and GM/WM boundaries, producing a unique intracortical intensity profile for each vertex. Furthermore, morphological mean curvature and cortical thickness features were generated using FastSurfer [69]. All these datasets are included in the data release [74].

**c. Superficial white matter surface-based sampling and feature mapping from in-vivo dataset.** SWM surfaces in the in-vivo dataset were sampled using the same methodology as for the *postmortem* histological datasets. Due to the lower resolution of the in-vivo dataset, SWM surfaces were sampled at 15 depths, each spaced 0.2 mm apart (Fig 2A). These SWM surfaces were subsequently mapped to each MRI. Similar to the histological datasets, voxel-wise quantitative features were interpolated using trilinear interpolation to all surface points along the newly mapped SWM surfaces. Finally, the measurements were registered to the Conte69 template surface (5k vertices per hemisphere). For the analysis of the association between neuroanatomy and MRI intensity in SWM, a linear regression model was used to correct SWM intensity for curvature effects. Regional variation in in-vivo microstructural profiles was assessed by computing central moments of intensity values sampled from cortex into the SWM.

### 3. Neural contextualization with cytoarchitecture

The SWM intensity profiles derived from histology and MRI were contextualized within the framework of normative variations in cortical cytoarchitecture, which detail the cellular composition and structural characteristics of cortical regions. These intensity profiles were categorized based on cortical types defined by Von Economo and Koskinas [30,96]. We also utilized the atlas proposed by Mesulam [97] to delineate four functional regions, following variations in cortical laminar differentiation. To map SWM to cortical types, the Laplacian field provided an isomorphic mapping between the cortex and underlying SWM, extending cortical labels along the surface vertices into the SWM.

## 4. Generation of SWM microstructural gradients

We adapted a nonlinear dimensionality reduction approach to derive eigenvectors that represent spatial gradients in SWM microstructural variation (Fig 4A). To reduce noise and enhance spatial continuity, intensity values from each MRI-derived intensity profile were smoothed separately for each surface. This was done using an iterative smoothing approach with a uniform kernel, where the smoothing process was repeated for 5 iterations with a relaxation factor of 0.5. Vertex-wise intensity profiles were then cross-correlated using Spearman's correlation. Partial correlations were applied to control for the average intensity profile, followed by log-transformation. This process produced microstructural profile covariance (MPC) matrices, capturing subject-specific similarities in myelin-related proxies across SWM depths. Each subject's MPC matrix was then transformed into a normalized angle affinity matrix and subjected to diffusion map embedding [98]. This nonlinear dimensionality reduction method has been shown to be resistant to noise in the input covariance matrix across various data types [24,71]. This approach revealed eigenvectors (or "gradients") that represent the primary spatial axes of variance in SWM microstructural profile similarities [99,100]. Such gradients provide a low-dimensional representation of SWM organization, highlighting macroscale patterns of microstructural variation. Gradient analyses were conducted using *BrainSpace* (http://github.com/MICA-MNI/BrainSpace) [101], applying the default sparsity setting (retaining the top 10% of MPC weights) and diffusion parameter ($\alpha = 0.5$) [101]. These settings were chosen based on previous studies [71], as they are known to preserve the global relationships between points in the embedded space. In this analysis, we concentrated on the principal gradient of microstructural similarity and computed subject-specific gradients. Spearman correlations were subsequently tested for the relationship between the SWM microstructural gradient and cortical geometry, as well as structural and functional connectivity properties. The analyses using the gradients were performed individually for each subject, and statistical significance was evaluated using spin permutation tests (1,000 permutations) implemented in *BrainSpace* [101]. We also applied a non-parametric locally weighted regression to model the relationship between the gradients and connectivity strengths at the group level, enabling a data-driven estimation of trends.

**Ethics statement:** The MRI data acquisition protocols were approved by the Research Ethics Board of McGill University and the Montreal Neurological Institute (REB #2023-8971 and REB #2022-8526). All participants provided written informed consent, including consent for open sharing of anonymized data.

---

Highlights

- Developed an anatomy-based open science framework to study SWM organization based on 3D *postmortem* histology and/or in-vivo high-field MRI

- The first 2 mm into the white matter are characterized by high structural complexity

- Inter-regional trends in SWM microstructural variation, which are related to cortical geometry, as well as cortico-cortical connectivity

---

## Supporting information

**S1 Fig. Alterations in intensity distributions of histological features across SWM depths, from the GM/WM interface to 3 mm into the SWM.**
(TIFF)

**S2 Fig. Spatial associations between alterations in intensity distributions and levels of laminar differentiation in histology and MRI features across SWM depths.**
(TIFF)

**S3 Fig. Spatial associations between alterations in intensity distributions and cytoarchitectural classes in histological and MRI features across SWM depths.**
(TIFF)

**S4 Fig. Variance explained by gradients of microstructural profile covariance from the in-vivo MRI datasets.** Scree plots show the eigenvalue spectrum for *(i)* T1 map and *(ii)* MTsat, illustrating the relative variance explained by each gradient. The data underlying this figure can be found in S1 Data.
(TIFF)

**S5 Fig. Variance explained by gradients of microstructural profile covariance from the histology datasets.** Scree plots display the eigenvalue spectra for *(i)* BigBrain and *(ii)* Ahead—Bielschowsky, and *(iii)* Ahead—Parvalbumin, illustrating the proportion variance explained by each gradient. The data underlying this figure can be found in S1 Data.
(TIFF)

**S6 Fig. Short-range structural connectivity (SC) correlations with SWM gradients. (A)** Connections with a geodesic distance <35 mm were selected to capture short fibers (*top* panel). Examples of three randomly selected vertices with geodesic distances <35 mm are shown in the *bottom* panel. **(B)** SC was restricted to this geodesic distance to isolate short-range connections (*top* panel). Short-range SC strength was calculated as the weighted sum of the short-range SC matrix across all nodes (*bottom* panel). **(C)** The *left* panel shows density plots of nonlinear correlations between connectivity strength and SWM gradients in the group analysis, with statistical significance assessed using spin permutation tests. The red dotted line represents the nonlinear fit capturing the relationship between gradients and connectivity strengths. The *right* panel presents subject-wise correlations between connectivity strength and SWM gradients derived from each qMRI. Box-and-whisker plots illustrate the distribution of Spearman correlation coefficients across the subjects, with nonsignificant correlations shown as light gray dots. The data underlying this figure can be found in S1 Data. **Abbreviations**: $r$, Spearman correlation; $P_{spin}$, Significance.
(TIFF)

**S7 Fig. (a)** The blockface image was downsampled to 0.8 mm resolution to generate a synthetic T1w image. **(b)** The GM/WM boundary was delineated on the downsampled blockface image. **(c)** The GM/WM surface and SWM surfaces were subsequently mapped onto the original histological stainings acquired at 0.2 mm isotropic resolution.
(TIFF)

**S1 Data. Supplementary tables.**
(XLSX)

**S1 Text. Abbreviations.**
(DOCX)

## Author contributions

**Conceptualization:** Youngeun Hwang, Raul Rodriguez-Cruces, Boris C. Bernhardt.

**Data curation:** Raul Rodriguez-Cruces, Donna Gift Cabalo.

**Formal analysis:** Youngeun Hwang.

**Investigation:** Youngeun Hwang, Donna Gift Cabalo.

**Methodology:** Youngeun Hwang, Raul Rodriguez-Cruces, Jordan DeKraker, Ilana R. Leppert, Risavarshni Thevakumaran, Christine L. Tardif, David A. Rudko, Casey Paquola, Pierre-Louis Bazin.

**Resources:** Donna Gift Cabalo, Pierre-Louis Bazin, Alan C. Evans.

**Software:** Raul Rodriguez-Cruces, Jordan DeKraker.

**Supervision:** Boris C. Bernhardt.

**Validation:** Youngeun Hwang.

**Visualization:** Youngeun Hwang.

**Writing – original draft:** Youngeun Hwang.

**Writing – review & editing:** David A. Rudko, Casey Paquola, Pierre-Louis Bazin, Andrea Bernasconi, Neda Bernasconi, Luis Concha, Boris C. Bernhardt.

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
