## [Editor Report · Decision Letter 0]

20 May 2025

Dear Dr Bernhardt,

Thank you for submitting your manuscript entitled "A unified imaging-histology framework for superficial white matter architecture studies in the human brain" for consideration by PLOS Biology.

Your manuscript has now been evaluated by the PLOS Biology editorial staff, as well as by an academic editor with relevant expertise, and I am writing to let you know that we would like to send your submission out for external peer review.

Important: We would like to consider your submission as a Resources & Methods - type article, rather than as a Research Article. When you fill out the next stage of the submission form, please select this article type.

Once your full submission is complete, your paper will undergo a series of checks in preparation for peer review. After your manuscript has passed the checks it will be sent out for review. To provide the metadata for your submission, please Login to Editorial Manager (https://www.editorialmanager.com/pbiology) within two working days, i.e. by May 22 2025 11:59PM.

Kind regards,

Taylor

Taylor Hart, PhD,

Associate Editor

PLOS Biology

thart@plos.org

---

## [Decision Letter · Decision Letter 1]

16 Jul 2025

Dear Dr Bernhardt,

Thank you for your patience while your manuscript "A unified imaging-histology framework for superficial white matter architecture studies in the human brain" was peer-reviewed at PLOS Biology. It has now been evaluated by the PLOS Biology editors, an Academic Editor with relevant expertise, and by several independent reviewers.

In light of the reviews, which you will find at the end of this email, we would like to invite you to revise the work to thoroughly address the reviewers' reports.

The reviewers say that the findings are novel and valuable. However, the reviewers raised several concerns about the study, including related to the 'unified framework', the integration of the histological and MRI data, and the analytic choices. They each pointed out several areas where the study would be strengthened by further quantification, additional analyses, and better integration of the different datasets. We would like to invite you to thoroughly address these points, through changes to the analyses and the text, as part of a Major Revision of your study. In particular, you should either provide a better data integration to justify your claims of employing a 'unified framework', or soften these claims.

Given the extent of revision needed, we cannot make a decision about publication until we have seen the revised manuscript and your response to the reviewers' comments. Your revised manuscript is likely to be sent for further evaluation by all or a subset of the reviewers.

**IMPORTANT - SUBMITTING YOUR REVISION**

*Re-submission Checklist*

*Published Peer Review*

*PLOS Data Policy*

*Blot and Gel Data Policy*

Sincerely,

Taylor

Taylor Hart, PhD,

Associate Editor

PLOS Biology

thart@plos.org

REVIEWS:

Reviewer #1: Summary:

The authors introduced a unified framework to study superficial white matter (SWM) organization based on 3D histological and high-field 7T MRI. Using the histology datasets, the authors first characterized depth-dependent variations in cellular staining, which were then analogously studied using in-vivo MRI measures including T1 relaxometry and magnetization transfer saturation. Then, they observed that the SWM principal gradient map derived from T1 exhibited significant spatial correlations with cortical geometry, as well as with structural and functional connectivity strength. The study thus provides novel insights into the organization of SWM in the human brain and underscores the potential of SWM mapping to advance fundamental and applied neuroscience research.

The topic is interesting and the open-science approach is commendable. The paper is also well-written. That being said, I have several comments, particularly related to, justifications of but the proposed "unified framework" is confusing due to insufficient justification for many analytical choices.

Major concerns:

The "unified framework" is a source of confusion as the relationship between the histological and MRI components is not clearly established. The analyses of histological data (Fig. 1) and MRI data (Figs. 2, 4, 5) are presented in parallel, but the manuscript lacks a direct comparison or validation to show how they are integrated. This makes it difficult to understand the contribution of the histology to the final conclusions, which are derived almost exclusively from the MRI data. We suggest the authors revise their description of the framework to more accurately reflect the workflow. For instance, framing it as a "histology-informed" or "conceptually-linked" approach, rather than a "unified" one, would better represent the study's design and make the findings clearer.

MTsat is generally considered more specific to myelin content than the T1 map, yet it shows weaker correlations with structural connectivity strength (Fig. 5B). How can this be explained? The authors should provide further discussion on this point.

In the Discussion (page 13, paragraph 2), the authors interpreted the correlation between the T1 map gradient and structural connectivity (SC) strength as reflecting the contribution of U-fibers. However, as SC was reconstructed using global tractography, short-range U-fibers are mixed with long-range association and projection fibers. It would be helpful if the authors selected only short streamlines and tested whether the correlations between SC and the T1 map gradient become stronger.

In Section 4, "Generation of SWM microstructural gradients," the rationale for using a diffusion map embedding approach should be clarified. Could the authors briefly explain what advantages this gradient-based analysis offers over more direct methods, such as analyzing SWM intensity values at specific depths? This would help readers better understand the motivation behind this key analytical step.

Minor Concerns:

In Section 4, "Generation of SWM microstructural gradients," the rationale for using a diffusion map embedding approach should be clarified. Could the authors briefly explain what advantages this gradient-based analysis offers over more direct methods, such as analyzing SWM intensity values at specific depths? This would help readers better understand the motivation behind this key analytical step.

Regarding the density maps in Figure 5B (middle panel), the visualization could be clarified. The plot shows high data density for structural connectivity strengths greater than 6, but the red trend line extends down to values around 4, where data appears to be sparse or absent. Please adjust the line to better reflect the range of the underlying data or clarify this in the caption.

Please report the variance explained by the principal gradients. A scree plot showing the eigenvalues for the leading gradients would be helpful to justify the focus on G1.

Please clarify the rationale for correlating SWM gradients with connectivity node strength. The principal gradient represents a global axis of microstructural organization, whereas node strength is a local, summary metric of connectivity. What is the underlying hypothesis that connects these two different scales of brain organization?

There appears to be a noteworthy contrast in the findings for MTsat and structural connectivity (Figure 5Bii). While the group-level correlation is reported as significant (P_spin = 0.024), the plot of individual correlations suggests the effect is non-significant in most subjects. Could the authors please comment on this? Clarifying whether this is due to a consistent but weak effect across subjects would help in the interpretation of the overall result.

Reviewer #2 [Basilis Zikopoulos]: The paper entitled "A unified imaging-histology framework for superficial white matter architecture studies in the human brain" is among the few studies that have tackled the superficial white matter (SWM) and a first effort to systematically study and compare SWM architecture across the entire cortex at multiple scales, using a multimodal approach that combined high resolution microscopy and histology with several high-field MRI datasets and in vivo scans. The authors analyzed both intensity profiles and statistical moments and the findings showed high structural complexity that varied across architectural gradients and showed variable correlations with structural and functional connectivity.

This is a very timely, interesting, and well written study and I want to commend the authors on their effort, care, and consideration of relevant literature, while completing this much-needed work. The strengths of this manuscript include (1) the impressive integration of two high-resolution histological microscopy datasets with several high-field 7T MRI and other in vivo structural, diffusion-weighted and functional MRI sets, (2) the rigorous approach used for SWM surface-based sampling that utilized the Laplacian vector field within the white matter domain, as well as approaches used to link systematic variations with cortical geometry and connectivity, and (3) the development of a very useful open access framework that can be widely used in future studies.

The findings highlight the intricate nature of SWM, are novel, and highly significant for understanding the relationship between architecture/structure, connectivity patterns, and function. They are a very important contribution not only to basic neuroscience but also for the future study of atypical function and pathology, and as such, the paper is of interest to both basic and clinical researchers. Importantly, the study's findings provide a foundation for future comparative studies, normative neuroimaging studies, and studies of white matter pathology in disorders.

Below few comments and suggestions that in my opinion will further increase the value and clarity of the manuscript, improve presentation, and strengthen reported findings and discussion:

Results and Methodological considerations

1. 50 fields were analyzed from the histological datasets; however, fewer were shown in Fig. 1B. Why not all? Please clarify and justify selection.

2. Similarly, 15 surfaces were analyzed from the MRI datasets however, Fig. 2B shows data from 16 surfaces. Which is it?

3. Results, Section 3, page 7: The authors state "The consistently low SD observed at the GM/WM interface across most of modalities likely reflects a shared anatomical transition across cortical areas, where similar intensity changes yield reduced regional variability". Which modalities did not show consistently low SD?

4. Results, Sections 4 and 5, pages 9 and 10 (associations and correlations): Since both MRI and histological datasets were eventually aligned to standard templates, why were these analyses only conducted on the MRI and not the histological datasets?

5. Materials and Methods, Section 1b, page 14: Since images were heavily preprocessed and downsampled, it would be useful to show example images in supplementary materials.

6. Materials and Methods, Section 1c, page 15: Clarifications and more details needed to describe the approach here for a more general readership. The authors state that the Laplacian field provided an isomorphic mapping between points on the SWM surface and the overlying cortex, enabling seamless integration of gray matter and white matter metrics. Is this how the surface-based sampling was eventually associated with different cortical regions? In other words, how did the authors extend the borders of grey matter regions to the SWM to define limbic, paralimbic, motor, etc. areas? This last comment also applies to the following Methods, section 3. Neural contextualization with cytoarchitecture, which lists Mesulam and vonEconomo cytoarchitectonic maps but the integration of these with the data is not clear.

7. Materials and Methods, Section 1c, page 15: in the same section, the authors state "The maximum SWM depth of 3 mm was chosen to capture both the U-fiber system and the termination zones of long-range bundles". Several sulcal areas in the cortex contain quite narrow white matter regions, such that the SWM is not deep (less than 3mm), often quickly changes to deep white matter and then SWM of a neighboring area/region. Similarly, subgenual cingulate areas (like posterior BA25), have very narrow SWM. Conversely, other cortical areas have a much thicker (deeper) SWM component. How was this tackled and taken this into account for the analysis and interpretation of the findings?

General and Discussion

8. In the Results, Section 3, bottom of page 7, the authors state "Together, these findings indicate that SWM is not a homogeneous region but a structured continuum, exhibiting depth-dependent microstructural variations shaped by cellular, fiber, and myelin organization, which vary across the brain". This finding is later discussed and there is a brief comparison with relevant data from other species. However, it should also be noted that this was independently shown for several WM regions across primates, based on density and distribution of axons and neurons, including cortical WM in chimpanzees (Swiegers et al., J. Comp. Neurol. 2021), cortical WM in rhesus macaques (Mortazavi et al., Frontiers in Neuroanatomy 2017 and2016), PFC WM in humans and non-human primates (Zikopoulos et al., PLOS Biology 2018).

9. In the Discussion, page 12, the authors state "Historically, the structural complexity of the SWM has challenged its neural characterization, with its cellular heterogeneity and diverse cytoarchitectonic features varying significantly across regions". The meaning of this sentence is not clear. Structural complexity, cellular heterogeneity and diverse cytoarchitectonic features vary significantly across gray and white matter areas. Complexity is a lot more pronounced in gray matter. Several studies have shown that systematic variability patterns in both gray and white matter can facilitate characterization of brain areas (e.g., classical vonEconomo or Sanides or Flechsig studies, and more recently Huntenburg et al., Cerebral Cortex 2017; Zikopoulos et al., PLOS Biology 2018).

10. In the Discussion, page 12, the authors state "These white matter interstitial neurons play a critical role during cortical development and have been reported to be associated with psychiatric disorders, where alterations in their density and distribution have been reported". A major disorder where axon and cell density and patterning are affected is autism (see Avino and Hutsler Brain Research 2010 and 2021).

11. In the Discussion, page 13, the authors state "Specifically, it showed a stronger correlation with structural connectivity, which may reflect the contribution of U-fibers, known to support local cortical communication and as a predominant feature of the SWM". This is in line with data showing major contribution of short-range connectivity in the WM,(see Rosen and Halgren PLOS Biology 2022).

Basilis Zikopoulos

Reviewer #3: This work provides a novel characterization of the superficial white matter using both histology and in-vivo MRI data. A surface-based approach was taken to derive the features for characterizing the depth-dependent variations of SWM properties. While this is very valuable work in improving our understanding SWM microstructures, the work mostly provide qualitative and global descriptions and lacks insight on SWM structure. More detailed comments are as follows:

1. The author claims a "framework" for studying SWM, but it is not clear what is the framework and how this framework is critical for studying SWM.

2. It's valuable to use the surface representation to study the SWM, but only global distributions were provided without providing much details of the local and regional variations of SWM.

3. Could the intensity profiles transitioning from GM to WM in Figure 1 and 2 be due to partial volume effects?

4. There is not much synergy in the results from histology and in-vivo MRI except both use the same surface and depth representation to extract SWM related features. Can the same gradient maps be computed for the histology features?

5. There is no insight on the principal gradient of the T1 map in Figure 4. Why does it depend on the sulcal/gyral regions? Is the sulcal/gyral separation quantitative or just qualitative observation? The association with cortical geometry is rather convenient but not fundamental.

6. Correlation with functional and structural connectivity is only based on global measures. There is a lack of more specific regional analyses. For structural connectivity, there is a lack of investigation of the impact of tractography techniques given the challenges in obtaining high quality U-fibers in SWM.

---

## [Decision Letter · Decision Letter 2]

4 Dec 2025

Dear Dr Bernhardt,

Thank you for your patience while we considered your revised manuscript "A computational framework to study superficial white matter architecture in the human brain on 3D histology and high-field MRI" for publication as a Research Article at PLOS Biology. This revised version of your manuscript has been evaluated by the PLOS Biology editors, the Academic Editor, and the original reviewers.

Based on the reviews, we are likely to accept this manuscript for publication. Please also make sure to address the following data and other policy-related requests.

IMPORTANT: Please ensure that you implement the following editorial points:

----------------

**Title:

-- We see that you have already modified your study's title in the latest revision. However, we have another suggestion. We think that emphasizing the potential for biological insights from your paper instead of the methodological aspects would be inviting to a broader set of readers. Is this alternative version acceptable to you?

"Linking microstructural features of the human superficial white matter architecture with cortical geometry and connectivity"

**Financial disclosure statement:

-- Please add links in the Financial Disclosure statement in the manuscript details.

**Ethics:

-- We acknowledge that your study uses previously-collected data. But as this data is from human subjects, please include in your ethics statement information about consent.

**Data:

-- As supplementary document files are not proofread, please upload your supplementary figures as independent files, and include the legends in the main text file.

-- Thank you for providing the data, code, and analysis notebooks. We also require the numerical data underlying the figures. Can you please include a supplemental data file, titled "S1 Data" and with the filename "S1_Data.xlsx" including the numerical data? You can upload this file in the supplement or include it in an online deposition.

This applies to the following figure panels:

4B

5AB (rightmost panels)

S4 (rightmost panels)

S5 (rightmost panels)

S6C (rightmost panels)

--- Please also cite the location of the data clearly in all relevant main and supplementary Figure legends, e.g. “The data underlying this Figure can be found in S1 Data” or “The data underlying this Figure can be found in https://doi.org/10.5281/zenodo.XXXXX”

**Code availability:

-- Thank you for providing the underlying code in GitHub. However, because Github depositions can be readily changed or deleted, please make a permanent DOI’d copy (e.g. in Zenodo) and provide this URL in the manuscript and Data Availability Statement.

----------------

We expect to receive your revised manuscript within two weeks.

*Published Peer Review History*

*Press*

Sincerely,

Taylor

Taylor Hart, PhD,

Associate Editor

thart@plos.org

PLOS Biology

REVIEWS

Reviewer #1 [Zaixu Cui]: Authors have addressed all my concerns. Thanks.

Reviewer #2 [Basilis Zikopoulos]: The revised version of the manuscript and the author responses addressed the points raised in the previous round of review satisfactorily.

Reviewer #3: I have no further question and would recommend the acceptance of this paper.

---

## [Editor Report · Decision Letter 3]

16 Jan 2026

Dear Dr Bernhardt,

Thank you for the submission of your revised Research Article "Microstructural profiles of the human superficial white matter and their associations to cortical geometry and connectivity" for publication in PLOS Biology. On behalf of my colleagues and the Academic Editor, Claus Hilgetag, I am pleased to say that we can in principle accept your manuscript for publication, provided you address any remaining formatting and reporting issues. These will be detailed in an email you should receive within 2-3 business days from our colleagues in the journal operations team; no action is required from you until then. Please note that we will not be able to formally accept your manuscript and schedule it for publication until you have completed any requested changes.

PRESS

Sincerely,

Taylor

Taylor Hart, PhD,

Associate Editor

PLOS Biology

thart@plos.org